# Empowering Our People: Syndemic Moderators and Effects of a Culturally Adapted, Evidence-Based Intervention for Sexual Risk Reduction among Native Americans with Binge Substance Use

**DOI:** 10.3390/ijerph19074283

**Published:** 2022-04-03

**Authors:** Christopher G. Kemp, Rachel Chambers, Francene Larzelere, Angelita Lee, Laura C. Pinal, Anna M. Slimp, Lauren Tingey

**Affiliations:** Center for American Indian Health, Johns Hopkins University, Baltimore, MD 21231, USA; rstrom3@jhu.edu (R.C.); flarzel1@jhu.edu (F.L.); aleebah1@jhu.edu (A.L.); lmelgar1@jhu.edu (L.C.P.); aslimp@jhu.edu (A.M.S.); ltingey1@jhu.edu (L.T.)

**Keywords:** binge substance use, sexual risk, Native American, indigenous, moderators

## Abstract

Native American (NA) communities are disproportionately affected by the intersecting, synergistic epidemics of sexually transmitted infections (STIs) and substance use. Targeted approaches to addressing these syndemics are critical given the relative scarcity of mental health and behavioral specialists in NA communities. We conducted a series of moderation analyses using data from a randomized controlled trial of the EMPWR (Educate, Motivate, Protect, Wellness, Respect) intervention for reducing sexual risk behaviors, culturally adapted for NA adults with recent binge substance use living on rural reservations. We considered several potential moderators and substance use and sexual risk outcomes at 6- and 12-months post-baseline. Three hundred and one people participated in the study. Age, marital status, educational attainment, employment, and depressive symptoms were differentially associated with intervention effects. EMPWR could be strengthened with the incorporation of additional skills-building related to condom use negotiation with casual partners. For individuals with lower educational attainment or without employment, additional supports and approaches to intervention may be needed. Importantly, this study did not identify intersecting sexual risk and substance use behaviors as moderators of EMPWR effectiveness, suggesting that NA adults with varying levels of risk behavior may be equally likely to benefit from this intervention.

## 1. Introduction

Native American (NA) communities are disproportionately affected by the intersecting, synergistic epidemics of sexually transmitted infections (STIs) and substance use, reflecting the intergenerational consequences of colonization and historical trauma, as well as ongoing racism and the chronic underfunding of critical health and social services for these communities [1,2,3]. The national reported rates of chlamydia and gonorrhea among American Indians and Alaska Natives were 760 per 100,000 and 366 per 100,000 in 2019, respectively, or approximately four and five times higher than among white adults [4]. An estimated 7.1% of American Indian and Alaska Native people over the age of 12 had an alcohol use disorder in 2018, which is 30% greater than the estimated prevalence among white people of the same age [5]. Alcohol and drug use have further been associated with increases in high-risk sexual behaviors in NA communities [6]. Recent research has highlighted the specific legacy of colonial violence—including the erosion of traditional practices, the introduction of alcohol, and the practice of boarding schools—that underlies these epidemiologic links [7]. As framed by the Indigenist Stress-Coping Model, NA values, spiritualities, and cultural practices promote the physical, social, and emotional health of NA people by buffering against these long-term effects of trauma and discrimination, including the dramatic disparities in physical and mental health [3,8,9,10,11]. However, many of these same values, spiritualties, and cultural practices have also been disrupted by the history of oppression, and there remains a strong need for culturally informed behavioral interventions to reduce the disproportionate effects of the synergistic epidemics of STIs and substance use in these communities [12]. 

Targeted approaches to delivering such interventions, focusing on the population subgroups most likely to benefit, are critical given the relative scarcity of mental health and behavioral specialists in NA communities [1,2,13,14]. Moderation analyses can serve to identify these subgroups. Prior analyses across diverse populations have found a range of demographic and behavioral characteristics that tend to moderate the effects of behavioral risk reduction interventions [13]. For example, baseline sexual risk, drug use, age, gender, and educational attainment have all been identified as moderators of the impact of HIV/STI or substance use risk reduction programs [15,16]. One meta-analysis of 174 HIV risk reduction studies found overall stronger effects on sexual risk among men, people engaged in sex work, and people who consume alcohol, while another pooled analysis identified weaker intervention effects among people who recently engaged in sex under the influence of drugs or alcohol [16,17]. A recent study found that participants with subsyndromal depression experienced greater reductions in sexual risk behaviors after receiving a risk reduction intervention, compared to participants with no or severe depression [18]. However, to our knowledge, no studies have identified the moderators of sexual risk reduction intervention effects among NA substance users. 

EMPWR (Educate, Motivate, Protect, Wellness, Respect) is a brief, client-focused counseling intervention for reducing sexual risk behaviors, adapted and culturally tailored for NA adults with recent binge substance use (including binge drinking or drug use), living on rural reservations, from the evidence-based Project RESPECT intervention [19]. Our study team recently completed a randomized controlled trial of EMPWR and found that intervention participants reported significantly fewer unprotected sex acts, were more likely to refuse sex if their partner was not tested, and were more likely to complete STI self-screening, compared to control participants [20]. The present study builds on this work to fill a gap in the literature regarding the moderators of sexual risk reduction effects across intersecting risk factors in NA communities. Our aim is to explore heterogeneity in EMPWR intervention effects as a function of intersecting syndemic factors at the participant level, including overlapping sexual risk behaviors and substance abuse. Results will inform future work to address disparities in NA health by identifying sub-populations at specific intersections of risk who are most likely to benefit from EMPWR and comparable interventions.

## 2. Materials and Methods

### 2.1. Study Design and Participants

The study was a randomized controlled trial conducted in partnership with a rural reservation community in the Southwest. The goal was to evaluate the EMPWR program for impacts on sexual and substance use behaviors among adults with recent binge substance use [20]. The study protocol was approved by the participating Tribal community’s governing bodies including the Tribal Health Council and Health Board as well as the appropriate Indian Health Services Institutional Review Board and University Institutional Review Board. The participating Tribal community’s governing bodies reviewed and approved this manuscript. Details on the study methods have been published elsewhere [21]. 

Participants were eligible for the EMPWR trial if they met the following criteria: (1) identify as Native American, (2) 18–55 years of age, (3) reside in the participating tribal community, (4) provide written informed consent, (5) engaged in sexual activity in the prior three months and (6) had at least one episode of binge substance use within the prior three months. Participants were screened and enrolled into the trial from July 2015 to January 2018.

### 2.2. Randomization and Masking

Baseline assessments were conducted with consented participants by a blinded outcomes assessor. Blinded study staff then assigned participants to receive the EMPWR intervention or control using a 1:1 allocation ratio and stratified randomization technique to ensure balance of study conditions across gender and age group (i.e., 18–29, 30–39, 40–49 and 50–55). Participants and their intervention providers were not blinded to treatment randomization, though outcomes assessors and analysts remained blinded to treatment assignment for the duration of the study. A total of 301 participants were randomized to receive either the intervention or control. 

### 2.3. Intervention

EMPWR was adapted from Project RESPECT, which was an evidence-based intervention targeting HIV testing uptake and client-centered risk reduction [19]. Project RESPECT used counseling with motivational interviewing techniques, rooted in the theory of reasoned action and social cognitive theory, to change key determinants of condom use such as self-efficacy and perceived norms [22]. Community advisory board meetings and gender-specific focus group discussions with at-risk NA adults were used to adapt Project RESPECT to meet the needs of the participating tribal community [21]. Key changes included the intervention setting (from clinic to community-based), the facilitator (from clinic-based STI counsellor to paraprofessional community health worker from the tribal community), the test (from clinic-based HIV testing to STI self-test), and the target population (from STI clinic patients to NA adults with substance use problems). Other cultural adaptations included the use of local language and contextually relevant examples and resources. The final product, EMPWR, consists of two 40-minute sessions, delivered one-on-one, two weeks apart. 

In session one, facilitators help participants understand their personal risk factors for STIs (i.e., substance use, mental and emotional health, sexual health, socioeconomic status, etc.) and develop a personalized risk-reduction plan. This plan is tailored to the individual’s values and priorities and is designed to be realistic and achievable. STI screening through self-administered urine sample collection (for *Neisseria gonorrhea*, *Chlamydia trachomatis*, and *Trichomonas vaginalis*) was offered at the end of the first session. In session two, facilitators present the STI test result and provide additional counseling to support client-initiated behavior change and future STI risk-reduction planning. All participants who tested positive for a STI received treatment from a Public Health Nurse at the local IHS hospital. 

Facilitators received one initial week-long in-person training in the EMPWR curriculum. Weekly role-playing with study supervisors was conducted for two months. At the end of the two months facilitators had to pass a comprehensive curriculum exam with a score ≥ 85%. Quality assurance and fidelity to the core elements and key characteristics of the EMPWR program was maintained by randomly selecting 25% of all completed sessions for audio or video recording. Recordings were reviewed by study supervisors and fidelity monitoring checklists were completed. Feedback was provided to facilitators according to fidelity monitoring checklist criteria and additional curriculum training was delivered as necessary.

### 2.4. Control Condition

The control condition consisted of receiving Optimized Standard Care (OSC): a referral to outpatient care (mental health and/or substance abuse treatment) plus educational pamphlets and provision of information on substance use, signs, and symptoms of an STI, and information about STI screening resources. STI screening through self-administered urine sample collection and treatment for those testing positive was also offered to the control group. OSC was delivered to both intervention and control groups so that between-group differences could be attributed to the EMPWR intervention. 

### 2.5. Data Collection and Follow-Up

Participants attended a baseline visit prior to randomization as well as follow-up data collection visits at three- and six-months post-baseline. 

### 2.6. Outcomes

We had four outcomes of interest. The first was substance use in the prior three months, defined as either binge drinking (5+ drinks for males and 4+ drinks for females) or non-marijuana drug use (methamphetamine, heroin, cocaine, inhalants [e.g., sniffing glue, aerosol], or the abuse of over the counter or prescription medicines). The second was the number of self-reported unprotected sex acts (i.e., sex without a condom, or where the condom broke or fell off, or where the condom was used or removed mid-act) in the prior three months. The third was higher-risk sexual behavior, defined as self-reporting at least one new sexual partner and any lack of condom use during sex in the prior three months. The fourth outcome was whether participants opted to complete the self-administered STI screening that was offered as part of the study.

### 2.7. Moderators

We assessed several hypothesized moderators of intervention effects, all measured at baseline. Sociodemographic moderators included age group (18–34 vs. 35–55 years), gender (men vs. women), marital status (married or cohabitating vs. otherwise), children (any vs. none), educational attainment (less than high school graduate vs. high school or GED completed), and current employment (not employed vs. employed). Baseline mental and behavioral health moderators included depressive symptoms in the prior seven days (scoring 6+ on items 9, 16, 17, 18, 35, and 50 of the Brief Symptom Inventory [23], vs. 0–5), lifetime experience of physical violence (ever vs. none), lifetime experience of sexual violence (ever vs. none), binge drinking (5+ drinks for males and 4+ drinks for females in the prior three months, vs. otherwise), marijuana use (any use in the prior three months vs. none), other drug use (any use of methamphetamine, heroin, cocaine, inhalants [e.g., sniffing glue, aerosol], or the abuse of over the counter or prescription medicines in the prior three months vs. none), higher-risk sexual behavior (new sexual partner in the prior three months and lack of condom use, vs. otherwise), and the combination of binge drinking or drug use and higher-risk sexual behavior (binge drinking or drug use, combined with a new sexual partner and lack of condom use in the prior three months, vs. otherwise). 

### 2.8. Statistical Analysis

We descriptively analyzed participant characteristics at baseline, stratified by treatment assignment. 

We used longitudinal regression models to estimate the effects of the intervention on the outcomes of interest over time. Risk differences (RDs) between the intervention and control groups at three and six months were estimated using Gaussian generalized estimating equations (GEE) with identity link (i.e., linear probability models). The model accounted for correlation within patients over time [24]. Treatment group (binary), time (dummy variables for each time point), and treatment × time interactions were included as covariates. 

We then conducted moderation analyses to identify heterogeneity in intervention effects across baseline participant characteristics. Each of the moderation analyses included a moderator variable and its interactions with treatment, time, and treatment × time. The magnitude and statistical significance (*p* <  0.05) of the treatment × time × moderator interactions were assessed at each timepoint. We excluded five models in which the moderator variable and the outcome variable overlap (e.g., baseline binge drinking or substance use as moderators for change in the likelihood of binge drinking or substance use).

As a sensitivity analysis, we used the Benjamini and Hochberg procedure to account for multiple testing assuming a maximum false discovery rate of 0.25 [25]. 

Complete case analysis was used for all inferential models. All analyses were performed in version 4.0.5 of R [26]. 

## 3. Results

Three hundred and one individuals participated in the study: 150 in the intervention arm and 151 in the control arm (Table 1). Participant sociodemographic characteristics were similar between groups. Slightly less than half of participants were aged 35–55 years (40.0%) or were women (46.5%). Just over half (50.5%) had a high school diploma or GED, while few (9.1%) were currently employed. Depressive symptoms (32.6%) and prior experiences of physical violence (31.3%) were common. Almost all (82.7%) reported recent binge drinking, while a minority (40.3%) reported marijuana use or other drug use (29.4%), higher-risk sexual behavior (26.2%), or the combination of binge drinking, other drug use, and higher-risk sexual behavior (23.3%). 

Table 2 summarizes the overall and moderated intervention effects on the four outcomes of interest at three- and six-months post-baseline. The moderation analyses included 51 total models assessing the association of the fourteen potential moderating variables with the four outcomes, minus the five models excluded due to overlap between baseline moderator and outcome. Across these, we identified six statistically significant moderator effects. Intervention participants who completed high school or had a GED had substantially reduced likelihood of binge drinking or other drug use at six months post-baseline compared to intervention participants who did not finish high school or a GED (risk difference [RD]: −28%, 95% confidence interval [95% CI]: −55%, −2%). Unmarried or not cohabitating intervention participants had more unprotected sex acts (β: 23.24, 95% CI: 1.19, 45.31), and were more likely to report high-risk sexual behavior over the last three months (RD: 31%, 95% CI: 1%, 61%) at six months post-baseline compared to married or cohabitating intervention participants. Intervention participants with depressive symptoms had fewer unprotected sex acts (β: −25.96, 95% CI: −49.4, −2.51) compared to those without depressive symptoms at six months post-baseline, but were also more likely to report high-risk sexual behavior (RD: 39%, 95% CI: 9%, 68%). Employed intervention participants were much less likely to report high-risk sexual behavior at three months post-baseline compared to unemployed participants (RD: −57%, 95% CI: −100%, −14%).

None of the moderation effects reported in Table 2 remained statistically significant after accounting for multiple testing.

## 4. Discussion

We conducted an exploratory moderation analysis using data from a randomized trial testing the effects of an intervention to reduce risk associated with STIs among NA adults with recent binge substance. Our goal was to investigate whether sociodemographic and intersecting behavioral characteristics measured at baseline were prospectively associated with changes in four outcomes related to substance use and sexual risk. In contrast to previous moderation studies, neither gender nor baseline substance use and sexual risk appeared to moderate subsequent intervention effects [17,27,28]. Overall, little evidence of moderation was identified, suggesting that the intervention was uniformly effective across most participant characteristics. However, we found that age, marital status, educational attainment, employment, and depressive symptoms were differentially associated with intervention effects. These results should be interpreted with caution given the relatively small sample size and because of the strong potential for false positive associations given multiple testing, though they suggest several areas for future inquiry and opportunities to target and strengthen the EMPWR intervention. 

We found that participants with less education were less likely to reduce their substance use, while unemployed participants were less likely to reduce high risk sexual behaviors. Both findings relate to the underlying social determinants of health and suggest that the historical and institutional inequities faced by NA communities act not only to reinforce the syndemics of sexual risk and substance use, but also to minimize the effects of interventions aimed at these syndemics [29]. There are numerous barriers to educational attainment and reliable employment in rural NA communities, including high poverty rates, under-funded and under-staffed schools, historical trauma associated with residential school programs, and few employment opportunities. Previous studies have identified strong associations between lack of educational attainment and subsequent substance abuse, and between unemployment and poor mental and behavioral health [30,31]. Participants facing these issues may be more entrenched in their substance use and sexual risk behavior, and therefore less likely to respond to a brief intervention like EMPWR. 

Contradictory findings were observed in participants with depressive symptoms, who reported fewer sexual partners but were nevertheless more likely to report unprotected sex with a new partner. Though a large meta-analysis found no association between negative mood and sexual risk behavior overall, previous studies have suggested that people with depression may have reduced assertive skills and risk reduction self-efficacy, both of which are important for negotiating condom use with partners, and individual changes in depressive status have been found to be associated with increased sexual risk taking [32,33,34]. It may be that EMPWR participants with depressive symptoms learned skills to negotiate condom use with current partners, resulting in fewer unprotected sex acts, but that these skills were not sufficient to change behaviors with new partners. Additional exploration and validation of these findings is needed. 

Finally, we observed that participants who were married or living with a partner had greater reductions in sexual risk behavior compared to those who were not married or cohabitating. This directly contrasts a prior meta-analysis that found that individuals in a monogamous relationship were less likely to respond to sexual risk reduction interventions [17]. One potential explanation is that EMPWR may have reinforced the importance of protecting against both STIs and unwanted pregnancies, therefore increasing condom use and reducing sexual risk behavior, particularly among participants in stable relationships. 

This study had several limitations. First, there is an increased risk of false positive findings (i.e., Type I error) with multiple moderation analyses, and no statistically significant results remained after adjusting for multiple testing. We emphasize that this analysis was exploratory and argue that the reported associations still have value given the relative lack of research dedicated to targeting risk reduction interventions for NA sub-populations. Second, there was suboptimal participant retention over the study period: participants were highly mobile and had multiple intersecting marginalized identities, meaning even the experienced research team from the participating community had challenges completing follow-up data collection. Though incomplete retention led to data missingness, this missingness was not found to be associated with any observed covariates, and thus complete case analysis was a defensible approach [20,35]. Third, the substance use and risk behavior variables were self-reported and subject to potential social desirability and recall bias; however, any potential bias due to self-report would be equivalent across both study arms given random treatment assignment and equivalent data collection procedures across each arm. Fourth, we dichotomized all moderator variables and three out of four outcome variables, meaning participants with disparate risk behaviors or psychosocial characteristics will have been pooled into single categories (e.g., individuals with daily drug use will have been combined with individuals with one-time drug use into the ‘any drug use’ category). Increasing the sensitivity of our moderator and outcome variables would have required the use of more categories, increasing the likelihood of unstable estimates because of limited statistical power; we opted to dichotomize and preserve statistical power for this exploratory analysis. We also acknowledge that by categorizing all the sexual behavior of participants reporting new sexual partners and sex without condoms in the prior three months as ‘higher-risk’, we may have misclassified those who opted not to use condoms after both partners tested negative for STIs. Finally, a three-month recall window was used for the substance use and sexual risk variables. More narrow recall periods (e.g., two weeks) are standard for several of these variables and may improve measure precision. However, a three-month recall window was used because the first post-intervention assessment happened at three months, and the intent was to capture any differences in participant behavior over that entire period. 

## 5. Conclusions

Our findings have implications for the future delivery of EMPWR and other comparable risk reduction interventions in NA communities. Specifically, EMPWR may be targeted towards the people most likely to benefit from it, including those who are married or cohabitating, employed, who have high school diploma or GED, or who have depressive symptoms. Our results also suggest areas where EMPWR could be strengthened, including the incorporation of additional skill-building related to condom use negotiation with casual partners. More formative work is needed to build intervention effectiveness for individuals not in stable relationships, or for individuals with lower educational attainment or without employment. Importantly, this study did not identify intersecting sexual risk and substance use behaviors as moderators of EMPWR effectiveness, suggesting that NA adults with varying levels of risk behavior would be equally likely to benefit from this intervention.

## Figures and Tables

**Table 1 ijerph-19-04283-t001:** Baseline characteristics of participants in a trial of an intervention to reduce sexually transmitted infection risks among Native Americans with binge substance use (*n* = 301).

	Overall	Control	Intervention	*p*	% Missing
*n*	301	151	150		
Age 35–55 years (%)	120 (40.0)	63 (42.0)	57 (38.0)	0.56	0.3
Women (%)	140 (46.5)	71 (47.0)	69 (46.0)	0.95	0
Not Married/Cohabitating (%)	178 (63.6)	85 (60.7)	93 (66.4)	0.39	7.0
Have Children (%)	226 (75.3)	114 (76.0)	112 (74.7)	0.89	0.3
High School Diploma or GED (%)	149 (50.5)	79 (54.1)	70 (47.0)	0.27	2.0
Currently Employed (%)	27 (9.1)	14 (9.5)	13 (8.7)	0.96	1.3
Depressive Symptoms (%)	91 (32.6)	43 (31.6)	48 (33.6)	0.83	7.3
Ever Experienced Physical Violence (%)	82 (31.3)	43 (32.8)	39 (29.8)	0.69	13.0
Ever Experienced Sexual Violence (%)	28 (11.6)	17 (13.9)	11 (9.2)	0.34	19.6
Binge Drinking in Prior Three Months (%)	215 (82.7)	103 (79.8)	112 (85.5)	0.3	13.6
Marijuana Use in Prior Three Months (%)	106 (40.3)	58 (43.6)	48 (36.9)	0.327	12.6
Other Drug Use in Prior Three Months (%)	68 (29.4)	38 (34.2)	30 (25.0)	0.16	23.3
New Partner and Unprotected Sex in Prior Three Months (%)	68 (26.2)	36 (28.1)	32 (24.2)	0.57	13.6
Binge Drinking or Other Drug Use, and New Partner and Unprotected Sex in Prior Three Months (%)	60 (23.3)	33 (25.8)	27 (20.9)	0.44	14.6

**Table 2 ijerph-19-04283-t002:** Primary and moderation model estimates of intervention effects among participants in a trial of an intervention to reduce sexually transmitted infection risks among Native Americans with binge substance use (*n* = 301).

	Binge Drinking or Other Drug Use		Unprotected Sex Acts		Higher Risk Sexual Behavior		STI Screening	
	3 m	6 m	3 m	6 m	3 m	6 m	3 m	6 m E *
	RD [95% CI]	RD [95% CI]	β [95% CI]	β [95% CI]	RD [95% CI]	RD [95% CI]	RD [95% CI]	RD [95% CI]
**Primary Effects**								
Control	--	--	--	--	--	--	--	--
Intervention	−0.03 [−0.15, 0.09]	0.04 [−0.09, 0.17]	−3.49 [−12.24, 5.25]	−6.25 [−15.73, 3.22]	0.05 [−0.1, 0.2]	0.04 [−0.11, 0.18]	--	**0.14 [0.02, 0.27]**
**Moderators**								
Age								
18−34 years	--	--	--	--	--	--	--	--
35−55 years	−0.03 [−0.25, 0.2]	−0.16 [−0.41, 0.09]	−6.19 [−24.34, 11.95]	−6.75 [−26.17, 12.68]	0.15 [−0.17, 0.46]	0.1 [−0.18, 0.37]	--	0.11 [−0.15, 0.36]
Gender								
Male	--	--	--	--	--	--	--	--
Female	0 [−0.24, 0.23]	−0.19 [−0.44, 0.06]	13.56 [−3.8, 30.92]	14.08 [−4.85, 33.01]	0.16 [−0.14, 0.45]	0.09 [−0.2, 0.38]	--	0.12 [−0.13, 0.38]
Marital Status								
Married or Cohabitating	--	--	--	--	--	--	--	--
Not Married or Cohabitating	−0.15 [−0.39, 0.08]	0.11 [−0.16, 0.38]	3.28 [−17.52, 24.09]	**23.24 [1.18, 45.31]**	−0.14 [−0.45, 0.17]	**0.31 [0.01, 0.61]**	--	0.1 [−0.17, 0.37]
Children								
None	--	--	--	--	--	--	--	--
Any	0.17 [−0.15, 0.49]	0.01 [−0.31, 0.32]	−8.7 [−30.72, 13.32]	−6.61 [−30.37, 17.15]	0.23 [−0.07, 0.53]	0.18 [−0.14, 0.5]	--	0.19 [−0.1, 0.47]
Education								
Less than high school	--	--	--	--	--	--	--	--
High school graduate or GED	−0.13 [−0.36, 0.11]	**−0.28 [−0.55, −0.02]**	−4.24 [−22.4, 13.91]	−9.46 [−28.46, 9.54]	0.13 [−0.16, 0.43]	0.25 [−0.04, 0.53]	--	0.17 [−0.09, 0.44]
Employment								
Not employed	--	--	--	--	--	--	--	--
Employed	−0.21 [−0.71, 0.29]	0.05 [−0.37, 0.48]	−0.66 [−37.81, 36.48]	10.38 [−16.02, 36.78]	**−0.57 [−1, −0.14]**	−0.18 [−0.68, 0.33]	--	−0.09 [−0.51, 0.34]
Depressive Symptoms								
None	--	--	--	--	--	--	--	--
Any	−0.22 [−0.45, 0]	−0.03 [−0.31, 0.24]	−1.99 [−22.43, 18.45]	**−25.96 [−49.4, −2.51]**	0.24 [−0.07, 0.55]	**0.39 [0.09, 0.68]**	--	0.1 [−0.2, 0.4]
Physical violence								
None	--	--	--	--	--	--	--	--
Ever	−0.09 [−0.37, 0.19]	0.12 [−0.15, 0.39]	0.23 [−21.4, 21.86]	8.33 [−14.94, 31.61]	−0.13 [−0.46, 0.21]	−0.18 [−0.5, 0.13]	--	0.17 [−0.13, 0.48]
Sexual violence								
None	--	--	--	--	--	--	--	--
Ever	−0.36 [−0.74, 0.02]	−0.31 [−0.71, 0.09]	−10.15 [−39.7, 19.4]	−11.85 [−49.33, 25.64]	−0.04 [−0.44, 0.35]	0.02 [−0.39, 0.42]	--	−0.16 [−0.63, 0.31]
Binge drinking								
No	--	--	--	--	--	--	--	--
Yes	--	--	−2.01 [−17.8, 13.78]	3.74 [−20.83, 28.3]	−0.05 [−0.43, 0.32]	0.01 [−0.35, 0.37]	--	0.01 [−0.38, 0.39]
Marijuana use								
No	--	--	--	--	--	--	--	--
Yes	0.09 [−0.16, 0.35]	0.1 [−0.17, 0.37]	−2.59 [−22.01, 16.83]	−5.02 [−26, 15.96]	0.06 [−0.25, 0.37]	0.13 [−0.18, 0.44]	--	0 [−0.29, 0.29]
Other drug use								
No	--	--	--	--	--	--	--	--
Yes	--	--	−17.14 [−39.28, 4.99]	−8.77 [−33.92, 16.39]	0.25 [−0.11, 0.61]	−0.01 [−0.39, 0.37]	--	−0.04 [−0.38, 0.3]
New partner and unprotected sex								
No	--	--	--	--	--	--	--	--
Yes	0.02 [−0.22, 0.26]	0.16 [−0.11, 0.43]	−3.58 [−24.96, 17.8]	−1.88 [−23.59, 19.83]	--	--	--	0.17 [−0.13, 0.48]
Binge drinking or other drug use, and new partner/unprotected sex								
No	--	--	--	--	--	--	--	--
Yes	--	--	−3.14 [−25.56, 19.28]	−5.24 [−26.8, 16.33]	--	--	--	0.2 [−0.12, 0.52]

Footnotes: * Comparison of intervention vs. control by six months. Bold text indicates statistically significant estimate (*p* < 0.05). Abbreviations: RD, risk difference. CI, confidence interval.

## Data Availability

The data presented in this study are available on request from the corresponding author. The data are not publicly available due to privacy and ethical concerns.

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
