# Peer review of "Empowering Our People: Syndemic Moderators and Effects of a Culturally Adapted, Evidence-Based Intervention for Sexual Risk Reduction among Native Americans with Binge Substance Use"

_ijerph, 2022, doi:10.3390/ijerph19074283_

Round 1
Reviewer 1 Report
The authors of “Empowering our people: syndemic moderators and effects of a culturally adapted, evidence-based intervention for sexual risk reduction among Native Americans with binge substance use” were very responsive to the initial feedback.
A number of questions remain that, when addressed, would strengthen the manuscript.
Page 1, Line 33: The authors reference “Alaska Natives,” and later (lines 110 and 115) reference “AN,” but that acronym is not defined. If the references in lines 110 and 115 are to Alaska Natives, please define that acronym at the initial introduction in line 33.
Page 1, Lines 34-35: Unfortunately, the way this sentence was edited leaves a sentence that does not make sense: “An estimated 7.1% of NA people over the age of twelve had an alcohol use disorder in 2018, which is 30%.” Please complete that sentence (e.g., 30% of ______).
Page 4, Lines 154-155: In the initial review, there were questions about how higher risk sexual behavior was defined. In this draft, it says, “higher-risk sexual behavior, defined as self-reporting at least one new sexual partner and any lack of condom use during sex in the prior three months.” Lack of condom use with a partner in which (a) other birth control is being used and (b) both have tested negative for STIs would not necessarily be considered higher-risk sexual behavior in practice, yet, it sounds like it would be deemed higher-risk for these purposes. In addition, one new sexual partner may not be inherently risky if the new partner is a new, committed, monogamous partner with whom STI status and birth control has been discussed. Any rationale for the reliance on these definitions (e.g., citing an established precedent in the literature) would be helpful for the reader.
Page 4, Line 171: Now that cannabis/marijuana is no longer mentioned in the paper, its omission certainly stands out. Recreational drugs are not defined, and would need to be. If cannabis is included as a “recreational drug,” that needs to be made clear. If cannabis is not accounted for at all, that needs to be added to the limitation section.
Reviewer 2 Report
The authors have addressed my previous concerns.
Author Response
Thank you.
Reviewer 3 Report
The manuscript ¨Empowering our people: syndemic moderators and effects of a culturally adapted, evidence-based intervention for sexual risk reduction among Native Americans with binge substance use¨ is an interesting study evaluation the impact of an intervention on risk reduction for sexually transmitted diseases among native americans.
The study provides interesting preliminary data and was done using robust statistical methodologies. Authors found no effect for the moderators they considered which suggests that the intervention can have community wide effects.
In general the manuscript is well written and seems based on solid data and robust statistical analyses.
However, a few details need to be corrected. For example, the description of the ethical clearance description does not include details about the date and number of the ethical permit (see lines 331-334)
Similarly in lines 123-124 the names of the sexually transmitted pathogens are not italicized.
Point 5) in line 92 is not clear, do you mean ¨had sexual activity ...¨
lines 34-36 are not clear. What do you want to mean by ¨which is 30%¨. The langauge is not clear.
Round 2
Reviewer 1 Report
The authors were very responsive to the last round of feedback. The only remaining suggestion is to rename "recreational drugs" -- on top of the fact that there's been a real push away from this term (e.g., instead of "recreational cannabis" or "recreational marijuana" states have moved to "non-medical" or "personal" or "adult use," especially since we don't talk about "recreational alcohol use"), it sounds like the category covers "inhalants" and would therefore better be captured by that term.
Thank you for all you're doing to impact health!
Author Response
Thank you for your valuable comments and feedback. We feel the manuscript is strengthened because of your suggestions.
We have edited 'recreational drugs' to read 'inhalants' in the Outcomes and Moderators sections of the Methods.
This manuscript is a resubmission of an earlier submission. The following is a list of the peer review reports and author responses from that submission.
Round 1
Reviewer 1 Report
Thank you for the opportunity to review, "Empowering our people: syndemic moderators and effects of a culturally adapted, evidence-based intervention for sexual risk reduction among Native Americans with binge substance use." This article describes an approach to addressing substance use and sexual behavior within the NA population and importantly focuses on acknowleding social and contextual factors including historical oppression and marginalization contribute to syndemics that increase risk and poor health outcomes.
Introduction
Lines 35-39 - I wanted to know more about how NA values have been disrupted by historical oppression. Can the authors provide 2-3 more sentences on how the disruption in those values led to increased substance use, sexual risk, or other maladaptive health behaviors? I've thought of it more from a minority stress perspective and the use of unhealthy behaviors (e.g. drinking) to modulate discrimination related stress, but perhaps there are unique factors for NA individuals/communities that can be mentioned here?
Line 49 - remove the word "while"
Line 59 - Project RESPECT needs a citation. Would also be helpful to have a brief explanation of the core principles of the intervention and how/what makes EMPWR a "culturally adapted" version
Materials and Methods
The description of intervention and control conditions was detailed nicely. What was the cultural adaptation process like? How did investigators modify Project RESPECT to create EMPWR? More detail would be helpful to contextualize the current study and its findings, but also for replication/reproducibility of science and/or in case others want to use a similar approach for a different subpopulation.
Within Table 1, why did authors choose to use "MSM" as a category, but not indicate other sexual behavior categories/sexual orientation like men who have sex with women, women who have sex with men, women who have sex with women, etc. I suggest removing unless there is a strong reason to include. I imagine it is because rates of STIs/HIV are higher in MSM, however, in this context it does not seem helpful to highlight this group if other groups are not provided for comparison.
Results
Table 2 is difficult to read formatted as is because the estimate and the confidence interval are on different lines. If possible, spacing and font should be adjusted so it is easier to read.
Discussion
The discussion is a commentary on the results right now. Significantly more effort needs to go into comparing the current study findings with existing literature among NA populations and/or the general population to contextualize their findings. Authors should cite work that has already been done in explaining the current study.
The finding with depression and sexual risk is perhaps not that surprising given that the relationship between depression and sexual behavior is somewhat more complicated than authors present. There are many articles that could be referenced, for a few see: Crepaz, N., & Marks, G. (2001). Are negative affective states associated with HIV sexual risk behaviors? A meta-analytic review. Health Psychology, 20(4), 291–299. https://doi.org/10.1037/0278-6133.20.4.291;
Colfax, Vittinghoff, Husnik, McKirnan, Buchbinder, Koblin, Celum, Chesney, Huang, Mayer,Bozeman, Judson, Bryant, Coates, the EXPLORE Study Team, Substance Use and Sexual Risk: A Participant- and Episode-level Analysis among a Cohort of Men Who Have Sex with Men, American Journal of Epidemiology, Volume 159, Issue 10, 15 May 2004, Pages 1002–1012, https://doi.org/10.1093/aje/kwh135
The finding about married versus unmarried individuals and use of condoms seems very unexpected. While I appreciate authors' attempt to explain this difference based on the fact that "condom negotiation" might be easier with a known partner than an unknown partner this is actually somewhat illogical and I do not believe there is literature to support this explanation. It is actually usually the other way around - longer partnerships use condoms less often and therefore it is more challenging to negotiate condom use (and often less necessary from a health perspective if partners are monogomous). One alternative possibility is that these couples might have been using condoms as a form of pregnancy prevention? Were data collected around hormonal or other contraceptive use? Did participants provide a "reason" for using condoms (e.g. prevent pregnancy, prevent STIs)?
Limitations - Authors mention there was substantial dropout over time. Why do they think that is? Can they say anything about how the intervention was "received" by participants? Was there not enough cultural tailoring? Did participants not think the targets of the intervention were important?
Conclusions
The conclusion that the content around condom negotiation with casual partners "could be strengthened" does little to acknowledge that the goal of most sexual health interventions is this! Monogomous partnerships are significantly less likely to result in STI/HIV outcomes than concurrent partnerships or casual partners. It sounds more likely like the intervention "missed the target" here and more formative work needs to be done before the intervention is trialed again. Still, I commend the authors for conducting work that attempts to meet the health needs of underserved and historically marginalized communities including Native Americans.
Reviewer 2 Report
The authors of “Empowering our people: Syndemic moderators and effects of a culturally adapted, evidence-based intervention for sexual risk reduction among Native Americans with binge substance use” have submitted a well-written manuscript that considered moderators of EMPWR, concluding that while the intervention was uniformly effective across most participant characteristics, there were some differences by education and employment status. The authors make suggestions for strengthening EMPWR, as well as including additional skills building, supports, or approaches to intervention that may be needed for some.
Although the authors employed a strong and sophisticated analytic approach, the variables and definitions they included were very limited in their utility. For example, heavy episodic drinking (or “binge” drinking), is typically defined as exceeding 5+ drinks for males or 4+ drinks for females over a two week period (as was the case for college students when initially described by Wechsler, et al. 1995) or simply 5+ at least once in the past two weeks (as is the case in the Monitoring the Future Study). Alternatively, the window expands to 5+/4+ in the past month in SAMHSA’s National Survey of Drug Use and Health. The authors define a binge for their moderator analysis as “5+ drinks for males and 4+ drinks for females in the past three months, vs. otherwise (lines 149-150).” The rationale for this broad three month definition is not provided, nor is the variability in this window considered (i.e., someone with one “binge” episode says “yes,” as does someone who “binges” daily, and these are both nevertheless considered in the same category per the definition). It is also challenging to describe a “recent binge” as something that could have happened 89 days earlier.
The same concerns exist for drug use, defined as “any in the past three months vs. none (lines 150-151).” Someone who uses cannabis once is included in the same category (any in the past three months) as someone who reports daily use of methamphetamine, but these are in no way comparable risk behaviors or patterns. Without a more sensitive measure of substance use or “binge” occasions, the meaningfulness of outcomes reliant on substance use comes into question. Much more information about measures is needed and the definition of substance use should be provided much earlier in the manuscript.
The BSI for depression does not clarify which version of the BSI is used (or if it was only the 6 items in the depression subscale). This, too, raises questions about how “depressive symptoms” were defined (any symptoms vs. none; lines 146-147), since this treats equally the person who said they were feeling lonely one time with the person who endorses thoughts of ending their life multiple times – these are very different symptoms.
Finally, and similarly, the way that risky sexual behaviors are defined (lines 135-137) should be clarified and possibly refined. Is “unprotected sex” sex without a condom or sex without any birth control method? If all of these unprotected sex acts are with a single, long-term partner then the risk associated with these acts might be lower than other acts. In addition, one new sexual partner (e.g., a new dating partner) does not the pose the same sexual risk as 5 new sexual partners, yet it is not clear how these are differentiated under "higher-risk sexual behavior."
More information about measures, particularly the baseline measures, used in this study would strengthen the manuscript, as would including a discussion of any limitations associated with these simple thresholds and measures (which are currently not addressed). Similarly, adding a sentence about what “Project RESPECT” was and who the facilitators of the EMPWR program were would be helpful for readers to know to support efforts to translate findings to practice.
Reviewer 3 Report
The authors conducted a study of syndemics related to sexually transmitted infections (STIs) and substance use among Native American communities. This is an important research question to explore. I recommend the following points to clarify to strengthen the paper:
- More in-depth discussion in the introduction on how racism/discrimination/colonization affects binge substance use among Native Americans. This is the most important point of revision (the brief mention of colonization is a good start but needs to be expanded upon).
- Some information on prevalence of STI and substance use in Native Americans (not just the comparison to white adults).
- More background on some of your theories/models (e.g., Theory of Reasoned Action, Social Cognitive Theory, and the Indigenist Stress-Coping Model).
- More specific information on missingness, such as percentage missing for key items.
- How outliers were measured and handled in the analysis.
- How multicollinearity was assessed.
- Why you believe "any potential bias due to self-report would be equivalent across both study arms."
- Policy implications related to your work.
